# Bayesian Optimization with Robust Bayesian Neural Networks

**Jost Tobias Springenberg    Aaron Klein    Stefan Falkner    Frank Hutter**
Department of Computer Science
University of Freiburg
{springj,kleinaa,sfalkner,fh}@cs.uni-freiburg.de

## Abstract

Bayesian optimization is a prominent method for optimizing expensive-to-evaluate black-box functions that is widely applied to tuning the hyperparameters of machine learning algorithms. Despite its successes, the prototypical Bayesian optimization approach – using Gaussian process models – does not scale well to either many hyperparameters or many function evaluations. Attacking this lack of scalability and flexibility is thus one of the key challenges of the field. We present a general approach for using flexible parametric models (neural networks) for Bayesian optimization, staying as close to a truly Bayesian treatment as possible. We obtain scalability through stochastic gradient Hamiltonian Monte Carlo, whose robustness we improve via a scale adaptation. Experiments including multi-task Bayesian optimization with 21 tasks, parallel optimization of deep neural networks and deep reinforcement learning show the power and flexibility of this approach.

## 1   Introduction

Hyperparameter optimization is crucial for obtaining good performance in many machine learning algorithms, such as support vector machines, deep neural networks, and deep reinforcement learning. The most prominent method for hyperparameter optimization is Bayesian optimization (BO) based on Gaussian processes (GPs), as e.g., implemented in the Spearmint system [1].

While GPs are the natural probabilistic models for BO, unfortunately, their complexity is cubic in the number of data points and they often do not gracefully scale to high dimensions [2]. Although alternative methods based on tree models [3, 4] or Bayesian linear regression using features from a neural network [5] exist, they obtain scalability by partially sacrificing a principled treatment of model uncertainties.

Here, we propose to use neural networks as a powerful and scalable parametric model, while staying as close to a truly Bayesian treatment as possible. Crucially, we aim to keep the well-calibrated uncertainty estimates of GPs since BO relies on them to accurately determine promising hyperparameters. To this end we derive a more robust variant of the recent stochastic gradient Hamiltonian Monte Carlo (SGHMC) method [6].

After providing background (Section 2), we make the following contributions: We derive a general formulation for both single-task and multi-task BO with Bayesian neural networks that leads to a robust, scalable, and parallel optimizer (Section 3). We derive a scale adaptation technique to substantially improve the robustness of stochastic gradient HMC (Section 4). Finally, using our method – which we dub **B**ayesian **O**ptimization with **Hami**ltonian Monte Carlo **A**rtificial **N**eural **N**etworks (BOHAMIANN) – we demonstrate state-of-the-art performance for a wide range of optimization tasks. This includes multi-task BO, parallel optimization of deep residual networks, and deep reinforcement learning. An implementation of our method can be found at `https://github.com/automl/RoBO`.

## 2 Background

### 2.1 Bayesian optimization for single and multiple tasks

Let $f : \mathcal{X} \to \mathbb{R}$ be an arbitrary function defined over a convex set $\mathcal{X} \subset \mathbb{R}^d$ that can be evaluated at $\mathbf{x} \in \mathcal{X}$, yielding noisy observations $y \sim \mathcal{N}(f(\mathbf{x}), \sigma_{\text{obs}}^2)$. We aim to find $\mathbf{x}^* \in \arg\min_{\mathbf{x} \in \mathcal{X}} f(\mathbf{x})$. To solve this problem, BO (see, e.g., Brochu et al. [7]) typically starts by observing the function at an initial design $\mathcal{D} = \{(\mathbf{x}_1, y_1), \ldots, (\mathbf{x}_I, y_I)\}$. BO then repeatedly executes the following steps: (1) fit a regression model $p(f \mid \mathcal{D})$ to the current data $\mathcal{D}$; (2) use $p(f \mid \mathcal{D})$ to select an input $\mathbf{x}_{t+1}$ at which to query $f$ by maximizing an acquisition function (which trades off exploration and exploitation); (3) observe $y_{t+1} \sim \mathcal{N}(f(\mathbf{x}_{t+1}), \sigma_{\text{obs}}^2)$ and add the result to the dataset: $\mathcal{D} := \mathcal{D} \cup \{\mathbf{x}_{t+1}, y_{t+1}\}$.

In the generalized case of *multi-task Bayesian optimization* [8], there are $K$ related black-box functions, $\mathcal{F} = \{f_1, \ldots, f_K\}$, each with the same domain $\mathcal{X}$; and, the goal is to find $\mathbf{x}^* \in \arg\min_{\mathbf{x} \in \mathcal{X}} f_t(\mathbf{x})$ for a given $t$.[1] In this case, the initial design is augmented with previous evaluations of the related functions. That is, $\mathcal{D} = \mathcal{D}_1 \cup \cdots \cup \mathcal{D}_K$ with $\mathcal{D}_k = \{(\mathbf{x}_1^k, y_1^k), \ldots, (\mathbf{x}_{n_k}^k, y_{n_k}^k)\}$, where $y_i^k \sim \mathcal{N}(f_k(\mathbf{x}_i^k), \sigma_{\text{obs}}^2)$ and $n_k = |\mathcal{D}_k|$ points have already been evaluated for function $f_k$. BO then requires a probabilistic model $p(f \mid \mathcal{D})$ over the $K$ functions, which can be used to transfer knowledge from related tasks to the target task $t$ (and thus reduce the required number of function evaluations on $t$).

A concrete instantiation of BO is obtained by specifying the acquisition function and the probabilistic model. As acquisition function, here, we will use the popular expected improvement (EI) criterion [9]; other commonly used options, such as UCB [10] could be directly applied. EI is defined as

$$\alpha_{\text{EI}}(\mathbf{x}; \mathcal{D}) = \sigma(f(\mathbf{x}) \mid \mathcal{D}) \left( \gamma(\mathbf{x}) \Phi(\gamma(\mathbf{x})) + \phi(\gamma(\mathbf{x})) \right), \text{with } \gamma(\mathbf{x}) = \frac{\hat{y} - \mu(f(\mathbf{x}) \mid \mathcal{D})}{\sigma(f(\mathbf{x}) \mid \mathcal{D})}, \quad (1)$$

where $\Phi(\cdot)$ and $\phi(\cdot)$ denote the cumulative distribution function and the probability density function of a standard normal distribution, respectively, and $\mu(f(\mathbf{x}) \mid \mathcal{D})$ and $\sigma(f(\mathbf{x}) \mid \mathcal{D})$ denote the posterior mean and standard deviation of our probabilistic model based on data $\mathcal{D}$. While the prototypical probabilistic model in BO is a GP [1], we will use a Bayesian neural network (BNN).

### 2.2 Bayesian methods for neural networks

The ability to combine the flexibility and scalability of (deep) neural networks with well-calibrated uncertainty estimates is highly desirable in many contexts. Not surprisingly, there thus exist many approaches for this problem, including early work on (non-scalable) Hamiltonian Monte Carlo [11], recent work on variational inference methods [12, 13] and expectation propagation [14], reinterpretations of dropout as approximate inference [15, 16], as well as stochastic gradient MCMC methods based on Hamiltonian Monte Carlo [6] and stochastic gradient Langevin MCMC [17].

While any of these methods could, in principle, be used for BO, we found most of them to result in suboptimal uncertainty estimates. Our preliminary experiments – presented in the supplementary material (Section B) – suggest these methods often conservatively estimate the uncertainty for points far away from the data, particularly when based on little training data. This is problematic for BO, which crucially relies on well-calibrated uncertainty estimates based on few function evaluations. One family of methods that consistently resulted in good uncertainty estimates in our tests were Hamiltonian Monte Carlo (HMC) methods, which we will thus use throughout this paper. Concretely, we will build on the scalable stochastic MCMC method from Chen et al. [6].

## 3 Bayesian optimization with Bayesian neural networks

We now formalize the Bayesian neural network regression model we use as the basis of our Bayesian optimization approach. Formally, under the assumption that the observed function values (conditioned on $\mathbf{x}$) are normally distributed (with unknown mean and variance), we start by defining our probabilistic function model as

$$p(f_t(\mathbf{x}) \mid \mathbf{x}, \theta) = \mathcal{N}(\hat{f}(\mathbf{x}, t; \theta_\mu), \theta_{\sigma^2}), \quad (2)$$

where $\theta = [\theta_\mu, \theta_{\sigma^2}]^T$, $\hat{f}(\mathbf{x}, t; \theta_\mu)$ is the output of a parametric model with parameters $\theta_\mu$, and where we assume a homoscedastic noise model with zero mean and variance $\theta_{\sigma^2}$.[2] A single-task model can trivially be obtained from this definition:

**Single-task model.** In the single-task setting we simply model the function mean $\hat{f}(\mathbf{x}, t; \theta_\mu) = h(\mathbf{x}; \theta_\mu)$ using a neural network, with output $h$ (i.e. $h$ implements a forward-pass).

**Multi-task model.** For the multi-task model we use a slightly adapted network architecture. As additional input, the network is provided with a task-specific embedding vector. That is, we have $\hat{f}(\mathbf{x}, t; \theta_\mu) = h\left([\mathbf{x}; \psi_t]^T, \theta_h\right)$, where $h(\cdot)$, again, denotes the output of the neural network (here with parameters $\theta_h$) and $\psi_t$ is the $t$-th row of an embedding matrix $\psi \in \mathbb{R}^{K \times L}$ (we choose $L = 5$ for our experiments). This embedding matrix is learned alongside all other parameters. Additionally, if information about the dataset (such as data-set size etc.) is available it can be appended to this embedding vector. The full vector of the network parameters then becomes $\theta_\mu = [\theta_h, \text{vec}(\psi)]$, where $\text{vec}(\cdot)$ denotes vectorization. Instead of using a learned embedding we could have chosen to represent the tasks through a one-out-of-$K$ encoding vector, which functionally would be equivalent but would induce a large number of additional parameters to be learned for large $K$. With these definitions, the joint probability of the model parameters and the observed data is then

$$p(\mathcal{D}, \theta) = p(\theta_\mu) p(\theta_{\sigma^2}) \prod_{k=1}^{K} \prod_{i=1}^{|\mathcal{D}_k|} \mathcal{N}(y_i^k | \hat{f}(\mathbf{x}_i^k, k; \theta_\mu), \theta_{\sigma^2}), \qquad (3)$$

where $p(\theta_\mu)$ and $p(\theta_{\sigma^2})$ are priors on the network parameters and on the variance, respectively.

For BO, we need to be able to compute the acquisition function at given candidate points $\mathbf{x}$. For this we require the predictive posterior $p(f_t(\mathbf{x})|\mathbf{x}, \mathcal{D})$ (marginalized over the model parameters $\theta$). Unfortunately, for our choice of modeling $f_t$ with a neural network, evaluating this posterior exactly is intractable. Let us, for now, assume that we can generate samples $\theta^i \sim p(\theta \mid \mathcal{D})$ from the posterior for the model parameters given the data; we will show how to do this with stochastic gradient Hamiltonian Monte Carlo (SGHMC) in Section 4. We can then use these samples to approximate the predictive posterior $p(f_t(\mathbf{x})|\mathbf{x}, \mathcal{D})$ as

$$p(f_t(\mathbf{x})|\mathbf{x}, D) = \int_\theta p(f_t(\mathbf{x}) \mid \mathbf{x}, \theta) p(\theta \mid D) d\theta \approx \frac{1}{M} \sum_{i=1}^{M} p(f_t(\mathbf{x}) \mid \mathbf{x}, \theta^i). \qquad (4)$$

Using the same samples $\theta^i \sim p(\theta \mid \mathcal{D})$, we make a Gaussian approximation to this predictive distribution to obtain mean and variance to compute the EI value in Equation (1):

$$\mu(f_t(\mathbf{x})|\mathcal{D}) = \frac{1}{M} \sum_{i=1}^{M} \hat{f}(\mathbf{x}, t; \theta_\mu^i), \quad \sigma^2(f(\mathbf{x})|\mathcal{D}) = \frac{1}{M} \sum_{i=1}^{M} \left(\hat{f}(\mathbf{x}, t; \theta_\mu^i) - \mu(f_t(\mathbf{x})|\mathcal{D})\right)^2 + \theta_{\sigma^2}^i.$$
$$(5)$$

Notably, we can compute partial derivatives of $\alpha_{EI}$ (with respect to $\mathbf{x}$) via backpropagation through all functions $\hat{f}(\mathbf{x}, t; \theta_\mu^i)$ which allows gradient-based maximization of the acquisition function.

We also extended this formulation to parallel asynchronous BO by sampling possible outcomes for currently-running function evaluations and using the acquisition function $\alpha_{\text{MCEI}}$ proposed by Snoek et al. [1]. Details are given in the supplementary material (Section A).

# 4  Robust stochastic gradient HMC via scale adaptation

In this section, we show how stochastic gradient Hamiltonian Monte Carlo (SGHMC) can be used to sample from the model defined by Equation (3). We first summarize the general formalism behind SGHMC [6] and then derive a more robust variant suitable for BO.

## 4.1  Stochastic gradient HMC

HMC introduces a set of auxiliary variables, $\mathbf{r}$, and then samples from the joint distribution

$$p(\theta, \mathbf{r} \mid \mathcal{D}) \propto \exp\left(-U(\theta) - \frac{1}{2}\mathbf{r}^T \mathbf{M}^{-1} \mathbf{r}\right), \qquad \text{with} \qquad U(\theta) = -\log p(\mathcal{D}, \theta) \qquad (6)$$

by simulating a fictitious physical system described by a set of differential equations, called Hamilton's equations. In this system, the negative log-likelihood $U(\theta)$ plays the role of a potential energy, $\mathbf{r}$ corresponds to the momentum of the system, and $\mathbf{M}$ represents the (arbitrary) mass matrix [18].

Classically, the dynamics for $\theta$ and $\mathbf{r}$ depend on the gradient $\nabla U(\theta)$ whose evaluation is too expensive for our purposes, since it would involve evaluating the model on all data-points. By introducing a user-defined *friction* matrix $\mathbf{C}$, Chen et al. [6] showed how Hamiltonian dynamics can be modified to sample from the correct distribution if only a noisy estimate $\nabla \tilde{U}(\theta)$, e.g. computed from a mini-batch, is available. In particular, their discretized system of equations reads

$$\Delta \theta = \epsilon \mathbf{M}^{-1} \mathbf{r}\,, \qquad\qquad \Delta \mathbf{r} = -\epsilon \nabla \tilde{U}(\theta) - \epsilon \mathbf{C} \mathbf{M}^{-1} \mathbf{r} + \mathcal{N}(0, 2(\mathbf{C} - \hat{\mathbf{B}})\epsilon)\,, \qquad (7)$$

where, in a suggestive notation, we write $\mathcal{N}(0, \Sigma)$ representing the addition of a sample from a multivariate Gaussian with zero mean and covariance matrix $\Sigma$. Besides the estimate for the noise of the gradient evaluation $\hat{\mathbf{B}}$, and an undefined step length $\epsilon$, all that is required for simulating the dynamics in Equation (7) is a mechanism for computing gradients of the log likelihood (and thus of our model) on small subsets (or batches) of the data. This makes SGHMC particularly appealing when working with large models and data-sets. Furthermore, Equation (7) can be seen as an MCMC analogue to stochastic gradient descent (with momentum) [6]. Following these update equations, the distribution of $(\theta, \mathbf{r})$ is the one in Equation (6), and $\theta$ is guaranteed to be distributed according to $p(\theta \mid \mathcal{D})$.

## 4.2 Scale adapted stochastic gradient HMC

Like many Monte Carlo methods, SGHMC does not come without caveats, namely the correct setting of the user-defined quantities: the friction term $\mathbf{C}$, the estimate of the gradient noise $\hat{\mathbf{B}}$, the mass matrix $\mathbf{M}$, the number of MCMC steps, and – most importantly – the step-size $\epsilon$. We found the friction term and the step-size to be highly model and data-set dependent[3], which is unacceptable for BO, where robust estimates are required across many different functions $\mathcal{F}$ with as few parameter choices as possible.

A closer look at Equation (7) shows why the step-size crucially impacts the robustness of SGHMC. For the popular choice $\mathbf{M} = \mathbf{I}$, the change in the momentum is proportional to the gradient. If the gradient elements are on vastly different scales (and potentially correlated), then the update effectively assigns unequal importance to changes in different parameters of the model. This, in turn, can lead to slow exploration of the target density. To correct for unequal parameter scales (and respect their correlation), we would ideally like to use $\mathbf{M}$ as a pre-conditioner, reflecting the metric underlying the model's parameters. This would lead to a stochastic gradient analogue of Riemann Manifold Hamiltonian Monte Carlo [19], which has been studied before by Ma et al. [20] and results in an algorithm called generalized stochastic gradient Riemann Hamiltonian Monte Carlo (gSGRHMC). Unfortunately, gSGRHMC requires computation (and storage) of the full Fisher information matrix of $U$ and its gradient, which is prohibitively expensive for our purposes.

As a pragmatic approach, we consider a pre-conditioning scheme increasing SGHMCs robustness with respect to $\epsilon$ and $\mathbf{C}$, while avoiding the costly computations of gSGRHMC. We want to note that recently – and directly related to our approach – adaptive pre-conditioning using ideas from SGD methods has been combined with stochastic gradient Langevin dynamics in Li et al. [21] and to derive a hybrid between SGD optimization and HMC sampling in Chen et al. [22]. These approaches however either come with additional hyperparameters that need to be set or do not guarantee unbiased sampling. The rest of this section shows how all remaining SGHMC parameters in our method are determined automatically.

**Choosing M.** For the mass matrix, we take inspiration from the connection between SGHMC and SGD. Specifically, the literature [23, 24] shows how normalizing the gradient by its magnitude (estimated over the whole dataset) improves the robustness of SGD. To perform the analogous operation in SGHMC, we propose to adapt the mass matrix *during the burn-in phase*. We set $\mathbf{M}^{-1} = \mathrm{diag}\left(\hat{V}_\theta^{-1/2}\right)$, where $\hat{V}_\theta$ is an estimate of the (element-wise) uncentered variance of the gradient: $\hat{V}_\theta \approx \mathbb{E}[(\nabla \tilde{U}(\theta))^2]$. We estimate $\hat{V}_\theta$ using an exponential moving average during the

burn-in phase yielding the update equation

$$\Delta\hat{V}_\theta = -\tau^{-1}\hat{V}_\theta + \tau^{-1}\nabla(\tilde{U}(\theta))^2, \tag{8}$$

where $\tau$ is a free parameter vector specifying the exponential averaging windows. Note that all multiplications above are element-wise and $\tau$ is a vector with the same dimensionality as $\theta$.

**Automatically choosing $\tau$.** To avoid adding $\tau$ as a new hyperparameter – that would have to be tuned – we automatically determine its value. For this purpose, we use an adaptive estimate previously derived for adaptive learning rate procedures for SGD [25]. We maintain an additional smoothed estimate of the gradient $g_\theta \approx \nabla U(\theta)$ and consider the element-wise ratio $g_\theta^2/\hat{v}_\theta$ between the squared estimated gradient and the gradient variance. This ratio will be large if the estimated gradient is large compared to the noise – in which case we can use a small averaging window – and it will be small if the noise is large compared to the average gradient – in which case we want a larger averaging window. We formalize these desiderata by simultaneously updating Equation 8,

$$\Delta\tau = -g_\theta^2\hat{V}_\theta^{-1}\tau + 1\,, \qquad \text{and} \qquad \Delta g_\theta = -\tau^{-1}g_\theta + \tau^{-1}\nabla\tilde{U}(\theta)\,. \tag{9}$$

**Estimating $\hat{\mathbf{B}}$.** While the above procedure removes the need to hand-tune $\mathbf{M}^{-1}$ (and will stabilize the method for different $\mathbf{C}$ and $\epsilon$), we have not yet defined an estimate for $\hat{\mathbf{B}}$. Ideally, $\hat{\mathbf{B}}$ should be the estimate of the empirical Fisher information matrix that, as discussed above, is too expensive to compute. We therefore resort to a diagonal approximation yielding $\hat{\mathbf{B}} = \frac{1}{2}\epsilon\hat{V}_\theta$ which is readily available from Equation (8).

**Scale adapted update equations.** Finally, we can combine all parameter estimates to formulate our automatically scale adapted SGHMC method. Following Chen et al. [6], we introduce the variable substitution $\mathbf{v} = \epsilon\mathbf{M}^{-1}\mathbf{r} = \epsilon\hat{V}_\theta^{-1/2}\mathbf{r}$ which leads us to the dynamical equations

$$\Delta\theta = \mathbf{v}\,, \qquad \Delta\mathbf{v} = -\epsilon^2\hat{V}_\theta^{-1/2}\nabla\tilde{U}(\theta) - \epsilon\hat{V}_\theta^{-1/2}\mathbf{C}\mathbf{v} + \mathcal{N}\left(0, 2\epsilon^3\hat{V}_\theta^{-1/2}\mathbf{C}\hat{V}_\theta^{-1/2} - \epsilon^4\mathbf{I}\right)\,, \tag{10}$$

using the quantities estimated in Equations (8)-(9) during the burn-in phase, and then fixing the choices for all parameters. Note that the approximation of $\hat{\mathbf{B}}$ cancels with the square of our estimate of $\mathbf{M}^{-1}$. In practice, we choose $\mathbf{C} = C\mathbf{I}$, i.e. the same independent noise for each element of $\theta$. In this case, Equation (10) constrains the choices of $C$ and $\epsilon$, as we need them to fulfill the relation $\min(V_\theta^{-1})C \geq \epsilon$. For the remainder of the paper, we fix $\epsilon = 10^{-2}$ (a robust choice in our experience) and chose $C$ such that we have $\epsilon\hat{V}_\theta^{-1/2}\mathbf{C} = 0.05\mathbf{I}$ (intuitively this corresponds to a constant decay in momentum of $0.05$ per time step) potentially increasing it to satisfy the mentioned constraint at the end of the burn-in phase.

We want to emphasize that our estimation/adaptation of the parameters only changes the HMC procedure during the burn-in phase. After it, when actual samples are recorded, all parameters stay fixed. In particular, this entails that as long as our choice of $\epsilon$ and $C$ satisfies $\min(\hat{V}_\theta^{-1})C \geq \epsilon$, our method samples from the correct distribution. Our choices are compatible with the constraints on the free parameters of the original SGHMC [6]. Further, we note that the scale adaptation technique is agnostic to the parametric form of the density we aim to sample from; and could therefore potentially also simplify SGHMC sampling for models beyond those considered in this paper.

## 5 Experiments on the effects of scale adptation

First, to test the efficacy of the proposed scale adaptation technique, we performed an evaluation on four common regression datasets following the protocol from Hernández-Lobato and Adams [14], presented in Table 1. The comparison shows that – despite its guarantees for sampling from the correct distribution – SGHMC (without our adaptation) required tuning for each dataset to obtain good uncertainty estimates. This effect can likely be attributed to the high dimensionality (and non-uniformity) of the parameter space (for which the standard SGHMC procedure might just require too many MCMC steps to sample from the target density). Our adaptation removed these problems. Additionally we found our method to faithfully represent model uncertainty even in regimes were only few data-points are available. This observation is qualitatively shown in Figure 1 (right) and further explored in the supplementary material.

Table 1: Log likelihood for regression benchmarks from the UCI repository. For comparison, we include results for VI (variational inference) and PBP (probabilistic backpropagation) taken from Hernández-Lobato and Adams [14]. We report mean ± standard deviation across 10 runs. The first two SGHMC variants are the vanilla algorithm (without our modifications) optimized for best mean performance (best average), and best performance on each dataset (tuned per dataset) via grid search.

| Method/Dataset | Boston Housing | Yacht Hydrodynamics | Concrete | Wine Quality Red |
|---|---|---|---|---|
| SGHMC (best average) | $-3.474 \pm 0.511$ | $-13.579 \pm 0.983$ | $-4.871 \pm 0.051$ | $-1.825 \pm 0.75$ |
| SGHMC (tuned per dataset) | $\mathbf{-2.489 \pm 0.151}$ | $-1.753 \pm 0.19$ | $-4.165 \pm 0.723$ | $-1.287 \pm 0.28$ |
| SGHMC (scale-adapted) | $-2.536 \pm 0.036$ | $\mathbf{-1.107 \pm 0.083}$ | $\mathbf{-3.384 \pm 0.24}$ | $\mathbf{-1.041 \pm 0.17}$ |
| VI | $-2.903 \pm 0.071$ | $-3.439 \pm 0.163$ | $-3.391 \pm 0.017$ | $-0.980 \pm 0.013$ |
| PBP | $-2.574 \pm 0.089$ | $-1.634 \pm 0.016$ | $\mathbf{-3.161 \pm 0.019}$ | $\mathbf{-0.968 \pm 0.014}$ |

## 6 Bayesian optimization experiments

We now show Bayesian optimization experiments for BOHAMIANN. Unless noted otherwise, we used a three layer neural network with 50 tanh units for all experiments. For the priors we let $p(\theta_\mu) = \mathcal{N}(0, \sigma_\mu^2)$ be normally distributed and placed a Gamma hyperprior on $\sigma_\mu^2$, which is periodically updated via Gibbs sampling. For $p(\theta_\sigma^2)$ we chose a log-normal prior. To approximate EI we used 50 samples acquired via SGHMC sampling. Maximization of the acquision function was performed via gradient ascent. Due to space constraints, full details on the experimental setup as well as the optimized hyperparameters for all experiments are given in the supplementary material (Section C), which also contains additional plots and evaluations for all experiments.

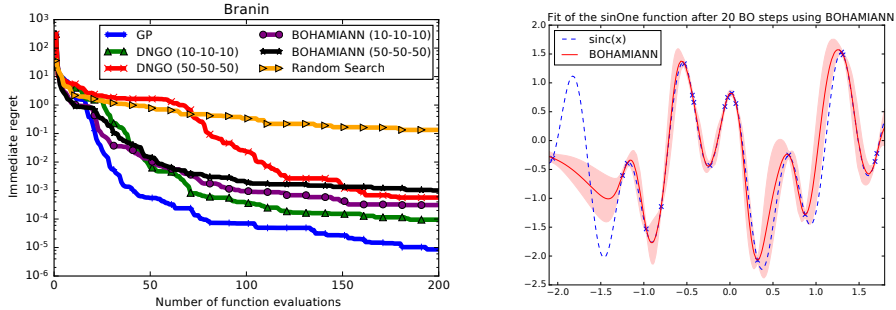

Figure 1: Evaluation on common benchmark problems. (Left) Immediate regret of various optimizers averaged over 30 runs on the Branin function. For DNGO and BOHAMIANN, we denote the layer sizes for the (3 layer) networks in parenthesis. (Right) A fit of the sinOne function after 20 steps of BO using BOHAMIANN. We plot the mean of the predictive posterior and ± 2 standard deviations; calculated based on 50 MCMC samples.

### 6.1 Common benchmark problems

As a first experiment, we compare BOHAMIANN to existing state-of-the-art BO on a set of synthetic functions and hyperparameter optimization tasks devised by Eggensperger et al. [2]. All optimizers achieved acceptable performance, but GP based methods were found to perform best on these low-dimensional benchmarks, which we thus take as a point of reference. Overall, on the 5 benchmarks BOHAMIANN matched the performance of GP based BO on 4 and performed worse on one, indicating that even in the low-data regime Bayesian neural networks (BNNs) are a feasible model class for BO. A detailed listing of the results is given in the supplementary material.

We further compared to our re-implementation of the recently proposed DNGO method [5], which uses features extracted from a maximum likelihood fit of a neural network as the basis for a Bayesian linear regression fit (and was also proposed as a replacement of GPs for scalable BO). For the benchmark tasks we found both DNGO and BOHAMIANN to perform well with BOHAMIANN being slightly more robust to different architecture choices. This behavior is illustrated in Figure 1 (left) where we compare DNGO with two different network architectures to BOHAMIANN.

Additionally, DNGO performed well for some high-dimensional problems (cf. Section 6.3), but it got stuck when we used it to optimize 13 hyperparameters of a Deep RL agent (cf. Section 6.4).

## 6.2 Multi-task hyperparameter optimization

Next, we evaluated BOHAMIANN for multi-task hyperparameter optimization of a support vector machine (SVM) and a random forest (RF) over a range of different benchmarks. Concretely, we considered a set of 21 different classification datasets downloaded from the OpenML repository [26]. These were grouped into four groups of related tasks (as determined by a distance based on metafeatures extracted from the datasets). Within each group (consisting of 3-6 datasets), we randomly designated the optimization of the algorithms hyperparameters for one dataset as the target function $f_t$. The remaining datasets were used for collecting $|\mathcal{D}_k| = 30$ additional training data points each, which were used as the initial design for BO. To allow for fast evaluation of this benchmark, we pre-computed the performance of different hyperparameter settings on all datasets following Feurer et al. [27]. The task for the optimizer then is to find an optimal hyperparameter setting for the target benchmark (for which it receives no initial data). We compared our method to the GP based multi-task BO procedure from Swersky et al. [8], as well as to standard, single-task, GP based BO. Overall, while all optimizers eventually found a solution close to the optimum the multi-task version of BOHAMIANN was able to exploit the knowledge obtained from the related datasets, resulting in quicker convergence. On average over all four benchmarks, MT-BOHAMIANN was *12 % faster* than GP based BO (to reach an immediate regret $\leq 0.25$), whereas MTBO was only $5\%$ faster. Plots showing the optimizer behavior are included in the supplementary material.

## 6.3 Parallel hyperparameter optimization for deep residual networks

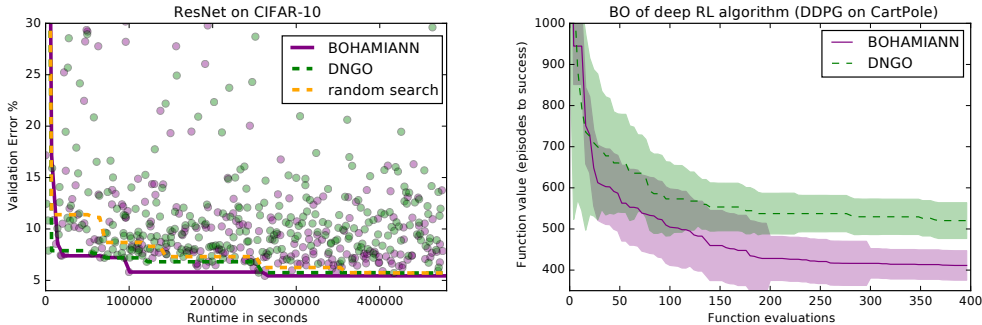

Figure 2: (Left) DNGO vs.BOHAMIANN for optimizing the 8 hyperparameters of a deep residual network on CIFAR-10; we plotted each function evaluation performed over time, as well as the current best; parallel random search is included as an additional baseline. (Right) DNGO vs. BOHAMIANN for optimizing the 12 hyperparameters of an RL agent.

Next, we optimized the hyperparameters of the recently proposed residual network (ResNet) architecture [28] for classification of CIFAR-10. We adopted a general parameterization of this architecture, tuning both the parameters of the stochastic gradient descent training as well as key architectural choices (such as the dimensionality reduction strategy used between residual blocks). We kept the maximum number of parameters fixed at the number used by the 32 layer ResNet [28]. Training a single ResNet took up to 6 hours in our experiments and we therefore used the parallel BO procedure described in Section 1 of the supplementary material (evaluating 8 ResNet configurations in parallel, for all of DNGO, random search, and BOHAMIANN).

Interestingly, all methods quickly found good configurations of the hyperparameters as shown in Figure 2(left), with BOHAMIANN reaching the validation performance of the manually-tuned baseline ResNet after 104 function evaluations (or approximately 27 hours of total training time). When re-training this model on the full dataset it obtained a classification error of **7.40 % ± 0.3**, matching the performance of the hand-tuned version from He et al. [28] (7.51 %). Perhaps surprisingly, this result was reached with a different architecture than the one presented in He et al. [28]: (1) it used max-pooling instead of strided convolutions for the spatial dimensionality reduction; (2) approximately $50\%$ of the weights in all residual blocks were shared (thus reducing the number of parameters).

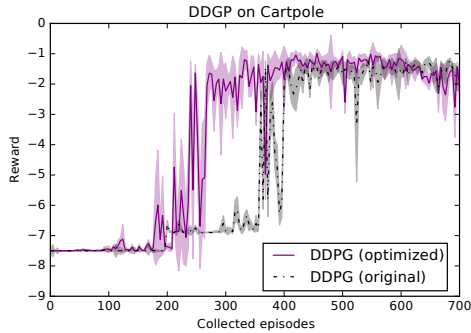

Figure 3: Learning curve for DDPG on the Cartpole benchmark. We compare the original hyperparameter settings to an optimized version of DDPG. The plot shows the cumulative reward (over 100 test episodes) obtained by the DDPG algorithm after it obtained $x$ episodes of data for training.

Table 2: Comparison between the original DDPG algorithm and a version optimized using BOHAMIANN on two control tasks. We show the number of episodes required to obtain successful performance in 10 consecutive test episodes (reward above -2 for CartPole, above -6 for reaching) and the maximum reward achieved by the controller.

| Cartpole | Reward | Episodes |
|---|---|---|
| DDPG | -1.18 | 470 |
| DDPG + DNGO | -1.39 | 507 |
| DDPG + BOHAMIANN | -1.46 | **405** |

| 2-link reaching task | Reward | Episodes |
|---|---|---|
| DDPG | -4.36 | 1512 |
| DDPG + DNGO | -4.39 | 1642 |
| DDPG + BOHAMIANN | -4.57 | **1102** |

## 6.4 Hyperparameter optimization for deep reinforcement learning

Finally, we optimized a neural reinforcement learning (RL) algorithm on two control tasks: the Cartpole swing-up task and a two link robot arm reaching task. We used a re-implementation of the DDPG algorithm by Lillicrap et al. [29] and aimed to minimize the interaction time with the simulated system required to achieve stable performance (defined as: solving the task in 10 consecutive test episodes). This is a critical performance metric for data-efficient RL.

The results of this experiment are given in Table 2 . While the original DDPG hyperparameters were set to achieve robust performance on a large set of benchmarks (and out-of-the-box DDPG performed remarkably well on the considered problems) our experiments indicate that the number of samples required to achieve good performance can be substantially reduced for individual tasks by hyperparameter optimization with BOHAMIANN. In contrast, DNGO did not perform as well on this specific task, getting stuck during optimization, see Figure 2 (right). A comparison between the learning curves of the original and the optimized DDPG, depicted in Figure 3, confirms this observation. The parameters that had the most influence on this improved performance were (perhaps unsurprisingly) the learning-rates of the Q-and policy networks and the number of SGD steps performed between collected episodes. This observation was already used by domain experts in a recent paper by Gu et al. [30] where they used 5 updates per sample (the hyperparameters found by our method correspond to 10 updates per sample).

## 7 Conclusion

We proposed BOHAMIANN, a scalable and flexible Bayesian optimization method. It natively supports multi-task optimization as well as parallel function evaluations, and scales to high dimensions and many function evaluations. At its heart lies Bayesian inference for neural networks via stochastic gradient Hamiltonian Monte Carlo, and we improved the robustness thereof by means of a scale adaptation technique. In future work, we plan to implement Freeze-Thaw Bayesian optimization [31] and Bayesian optimization across dataset sizes [32] in our framework, since both of these generate many cheap function evaluations and thus reach the scalability limit of GPs. We thereby expect substantial speedups in the practical hyperparameter optimization for ML algorithms on big datasets.

## Acknowledgements

This work has partly been supported by the European Commission under Grant no. H2020-ICT-645403-ROBDREAM, by the German Research Foundation (DFG), under Priority Programme Autonomous Learning (SPP 1527, grant HU 1900/3-1), under Emmy Noether grant HU 1900/2-1, and under the BrainLinks-BrainTools Cluster of Excellence (grant number EXC 1086).

## Footnotes

[1]The standard single-task case is recovered when $K = t = 1$.

[2]We note that, if required, we could model heteroscedastic functions by defining the observation noise variance $\theta_{\sigma^2}$ as a deterministic function of $\mathbf{x}$ (e.g. as the second output of the neural network).

[3]We refer to Section 5 for a quantitative evaluation of this claim.

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
