[Supplementary Material · bohamiann_nips16_supplementary.pdf]

# Supplementary material for the paper:
# Bayesian Optimization with
# Robust Bayesian Neural Networks

**Jost Tobias Springenberg  Aaron Klein  Stefan Falkner  Frank Hutter**
Department of Computer Science
University of Freiburg
{springj,kleinaa,sfalkner,fh}@cs.uni-freiburg.de

## Abstract

In this supplementary material, we provide additional details for the experimental setup of the paper: Bayesian Optimization with Robust Bayesian Neural Networks. We also present a set of additional plots for each experiment from the main paper.

## A   Extension to parallel Bayesian optimization

In this section, we define a variant of our algorithm for settings in which we can perform multiple evaluations of $f_t$ in parallel. Utilizing such parallel (and asynchronous) function evaluations in a principled manner for BO is non-trivial as we ideally would like to marginalize over the outcomes of currently running evaluations when suggesting new parameters $\mathbf{x}$ for which we want to query $f_t$. To achieve this, Snoek et al. [1] proposed an acquisition function which we refer to as Monte Carlo EI $\alpha_{\mathrm{MCEI}}$ that we adopt here for our model. Formally, we estimate $\alpha_{\mathrm{MCEI}}$ as

$$
\begin{aligned}
\alpha_{\mathrm{MCEI}}(\mathbf{x}; \mathcal{D}, R) &= \int_{\theta, y} \alpha_{\mathrm{EI}}\left(\mathbf{x}; \mathcal{D} \cup \{(\mathbf{x}_i, y_i^t)\}_{\mathbf{x}_i \in R}\right) p(y_i^t \mid \mathbf{x}_i, \theta) p(\theta \mid \mathcal{D}) d\theta dy \\
&\approx \frac{1}{M} \sum_{k=1}^{M} \alpha_{\mathrm{EI}}\left(\mathbf{x}; \mathcal{D} \cup \{(\mathbf{x}_i, {}^k y_i^t)\}_{\mathbf{x}_i \in R}\right) \\
\text{with} \quad {}^k y_i^t &\sim p(y_i^t \mid \mathbf{x}_i, \theta) p(\theta \mid \mathcal{D}),
\end{aligned}
\tag{1}
$$

where $\{(\mathbf{x}_i, {}^k y_i^t)\}_{\mathbf{x}_i \in R}$ is the set of currently running function evaluations (and their predicted targets) and where, in practice, we simply take $M = 1$ sample. Note that the computation of $\alpha_{EI}$ within the sum again requires multiple samples from the HMC procedure. As for EI we can differentiate through the computation of Equation (1) and maximize it using gradient ascent.

## B   Computational Requirements

For any BO method it is important to keep the computational requirements for training and evaluating the model in mind. To this end we want to draw a comparison between our method and DNGO with respect to computational costs. First, SGHMC sampling is similarly cheap as standard SGD training of neural networks, i.e. training a DNGO model from scratch and sampling via SGHMC has similar computational costs. If we were to start sampling from scratch with every incoming data-point and would fix the number of MCMC steps to be equivalent to K runs through the whole dataset then the computational complexity for sampling would grow linearly with the number of data-points.

In practice, we warm-start the optimizer during BO with the last sample from the previous evaluation and perform 2000 SGHMC steps (corresponding to 2000 batches) as burn-in, followed by $50 \cdot 50$

Figure 1: Four fits of the sinc function from 20 data-points. On the top-left the regression task was solved using our re-implementation of the Bayes by Backprop (BBB) approach from Blundell et al. [2]. On the top-right we used our re-implementation of the Dropout MC approach from Gal and Ghahramani [3]. In the bottom-left probabilistic Backpropagation [4] was used. On the bottom-right is a fit using SGHMC. As it can be observed most methods are overly confident, or have constant uncertainty bands, in large regions of the input space. Note that this function has no observation noise.

sampling steps (retaining every 50th sample). This budget was fixed for all tasks. SGHMC sampling is thus slightly faster than DNGO and orders of magnitude faster than GPs (see also the comparison between DNGO and GPs from Snoek et al. [5]); including acquisition function optimization it takes < 30 seconds. With more function evaluations we would increase the budget but expect a runtime of < 2min to select the next point for 50k function evaluations. We note that, if one wants to perform BO in large input spaces (e.g. for ML models with a very large number of parameters) it could be necessary to also increase the size of the used neural network model.

## C  Additional Experiments

### C.1  Obtaining well calibrated uncertainty estimates with Bayesian neural networks

As mentioned in the main paper, there exists a large body of work on Bayesian methods for neural networks. In preliminary experiments, we tried several of these methods to determine which algorithm was capable of providing well calibrated uncertainty estimates. All approximate inference methods we looked at (except for the MCMC variants) exhibited one of two problems (including the variational inference method from Blundell et al. [2], the method from Gal and Ghahramani [3] as well as the expectation propagation based approach from Hernández-Lobato and Adams [4]): either they did severely underfit the data, or they poorly predicted the uncertainty in regions far from observed data points. The latter behaviour is exemplified in Figure 1 (left) where we regressed the sinc function from 20 observations with a two layer neural network (50 tanh units each) using our implementation of the Bayes by Backprop (BBB) aproach from Blundell et al. [2]. In contrast, a fit of the same data with our method more faithfully represents model uncertainty as depicted in Figure 1 (right).

Table 1: Log likelihood for regression benchmarks from the UCI repository. For comparison we include results for VI (variational inference) and PBP (probabilistic backpropagation) for Bayesian neural network training taken from Hernández-Lobato and Adams [4]. We report mean $\pm$ stddev across 10 runs.

| Method/Dataset | Boston Housing | Yacht Hydrodynamics | Concrete | Wine Quality Red |
|---|---|---|---|---|
| SGHMC (best average) | -3.474 $\pm$ 0.511 | -13.579 $\pm$ 0.983 | -4.871 $\pm$ 0.051 | -1.825 $\pm$ 0.75 |
| SGHMC (tuned per dataset) | **-2.489 $\pm$ 0.151** | -1.753 $\pm$ 0.19 | -4.165 $\pm$ 0.723 | -1.287 $\pm$ 0.28 |
| SGHMC (scale-adapted) | -2.536 $\pm$ 0.036 | **-1.107 $\pm$ 0.083** | **-3.384 $\pm$ 0.24** | **-1.041 $\pm$ 0.17** |
| VI | -2.903 $\pm$ 0.071 | -3.439 $\pm$ 0.163 | -3.391 $\pm$ 0.017 | -0.980 $\pm$ 0.013 |
| PBP | -2.574 $\pm$ 0.089 | -1.634 $\pm$ 0.016 | **-3.161 $\pm$ 0.019** | **-0.968 $\pm$ 0.014** |

Table 2: Root mean squared error (RMSE) for regression benchmarks from the UCI repository. For comparison we include results for VI (variational inference) and PBP (probabilistic backpropagation) for Bayesian neural network training taken from Hernández-Lobato and Adams [4]. We report mean $\pm$ stddev across 10 runs.

| Method/Dataset | Boston Housing | Yacht Hydrodynamics | Concrete | Wine Quality Red |
|---|---|---|---|---|
| SGHMC (best average) | 3.719 $\pm$ 0.588 | 3.057 $\pm$ 0.05 | 9.505 $\pm$ 1.08 | 1.328 $\pm$ 0.61 |
| SGHMC (tuned per dataset) | **3.179 $\pm$ 0.435** | 1.594 $\pm$ 0.45 | **6.7136 $\pm$ 1.59** | **0.655 $\pm$ 0.281** |
| SGHMC (scale-adapted) | **3.107 $\pm$ 0.21** | **0.753 $\pm$ 0.17** | **6.325 $\pm$ 0.82** | 0.696 $\pm$ 0.062 |
| VI | 4.320 $\pm$ 0.2914 | 6.887 $\pm$ 0.6749 | 7.128 $\pm$ 0.1230 | 0.646 $\pm$ 0.0081 |
| PBP | **3.014 $\pm$ 0.1800** | 1.015 $\pm$ 0.0542 | **5.667 $\pm$ 0.0933** | **0.635 $\pm$ 0.0079** |

## C.2 Comparison between scale adaptive SGHMC and standard SGHMC

To quantify the influence of the scale adaption technique for SGHMC introduced in the main paper, we performed a set of additional experiments on four commonly used regression data-sets from the UCI repository. The data-sets were randomly split into a training and validation set (we used $10\%$ of the data for validation). We used the single function model from Section 2 of the main paper. The network consisted of a single hidden layer of 50 units, to simplify comparison to results from the literature. We compare SGHMC with and without our modification. For the non scale adaptive SGHMC variant, we performed a coarse grid-search over the relevant hyperparameters. We used the ranges $\epsilon \in [10^{-8}, 10^{-4}]$, $C \in [10^{-3}, 10^{-1}]$, chose $\hat{B} = 0$ and fixed the number of training steps to 15000. For the scale adaptive variant we used the same parameters as in the main paper $\epsilon = 10^{-4}$, $C = 0.05\mathbf{I}$ and a fixed number of steps (15000). A batch size of 32 was chosen for both methods. The results of this comparison are listed in Table 1 and Table 2, we also list results for standard SGHMC using the parameters that resulted in the best performance (on average) over all four data-sets. As can be observed for SGHMC good performance on any given data-set requires adaptation of the parameters whereas our method automatically stabilizes the performance.

# D Details for the experiments from the main paper

## D.1 Details on the experimental setup

For both the single task and the multi-task BO experiments, we use a three layer neural network with 50 units each (and tanh activation functions) to model the function mean $\hat{f}(\mathbf{x}, t; \theta_\mu)$ and let $\sigma$ be an additional scalar parameter. We note that although the network architecture is fixed it, obviously, has an influence on the expressive power of our model. However, we found that our method is relatively insensitive to the network definition due to the robustness of the MCMC procedure which also automatically tunes the model regularization. Specifically, we choose the prior on the network weights to be a normal distribution $p(\theta_\mu) = \mathcal{N}(0, \sigma_\mu^2)$ (equivalent to a weight decay in standard neural networks) and place a gamma hyperprior on its inverse variance, $p(\sigma_\mu^{2^{-1}}) = \mathrm{Gam}(\alpha, \beta)$ with $\alpha = \beta = 100$[1]. Updates to $\sigma_\mu^2$ are then performed via Gibbs sampling (every 100 HMC steps). For the prior on the variance of the noise model $p(\theta_{\sigma^2})$, we chose a log normal prior with variance

Figure 2: Immediate regret after each iteration of Gaussian processes, DNGO, random search and BOHAMIANN on 4 different synthetic functions.We report the median performance over 30 independent runs.

0.01. To obtain samples from $p(\theta \mid \mathcal{D})$, we use one persistent Markov chain during the complete experiment. Whenever a new data-point is acquired it is added to $\mathcal{D}$ and we perform 2000 burn-in steps, after which we take 1 sample every 50 steps until we have collected a total of 50 MCMC samples (i.e. 50 independent network parameters) that are then used for approximating the acquisition function. The step-size was set to $\epsilon = 10^{-2}$ for all experiments. To optimize the acquisition function for new target points to evaluate, we used gradient based optimization using the ADAM method [6] with default parameters. We highlight that it was not necessary to tune any of these parameters for good performance.

### D.2 Detailed results on common benchmark problems

Detailed results for BOHAMIANN on the benchmarks from Eggensperger et al. [7] in comparison to the state of the art from the literature are listed in Table 3. Except for the Logistic regression experiment, all methods eventually converged to a solution close to the optimum, as mentioned in the main paper. The table also shows results for our implementation of DNGO (using the same architecture as SGHMC). As can be observed, when using our code-base, we were unable to exactly match the performance reported in Snoek et al. [5] (a discrepancy exists for the logistic regression experiment and the LDA experiment). This discrepancy can be attributed to two factors: (1) Snoek et al. [5] placed a quadratic prior on the function mean (assuming that a functions' minimum is unlikely to be at the boundaries of the function) which we did not use in any of our experiments, (2) The DNGO hyperparameters were originally optimized over a set of unknown datasets and were not completely listed in the original paper.

Additional plots for the synthetic functions are given in Figure 2. After each iteration, we save the best observed point (incumbent) $\mathbf{x}_{inc}$ and plot the immediate regret $|f(\mathbf{x}_{inc}) - f(\mathbf{x}_\star)|$. These plots illustrate that while both DNGO and BOHAMIANN perform well (and are only slightly outperformed by GPs) DNGO is slightly less robust wrt. different parameter settings. Using the default network architecture from Snoek et al. [5] resulted in slowed convergence. In contrast, and reducing the network size to 10 units in each layer resulted in improved performance for DNGO. In comparison BOHAMIANN more robustly performed well for different network architectures.

Table 3: Experiments on benchmarks from the hyperparameter optimization library (HPOlib [7])

| Method | Branin(0.398) | Hartmann6(-3.322) | Logistic Regression | LDA (On grid) | SVM (On grid) |
|---|---|---|---|---|---|
| #Evals | 200 | 200 | 100 | 50 | 100 |
| SMAC | $0.655 \pm 0.27$ | $-2.977 \pm 0.11$ | $8.6 \pm 0.9$ | $1269.6 \pm 2.9$ | $24.1 \pm 0.1$ |
| TPE | $0.526 \pm 0.13$ | $-2.823 \pm 0.18$ | $8.2 \pm 0.6$ | $1271.5 \pm 3.5$ | $24.2 \pm 0.0$ |
| Spearmint | $0.398 \pm 0.00$ | $-3.133 \pm 0.41$ | $7.3 \pm 0.2$ | $1272.6 \pm 10.3$ | $24.6 \pm 0.9$ |
| Spearmint+Warp | $0.398 \pm 0.00$ | $-3.3166 \pm 0.02$ | $6.88 \pm 0.0$ | $1266.2 \pm 0.1$ | $24.1 \pm 0.1$ |
| DNGO | $0.398 \pm 0.00$ | $-3.319 \pm 0.02$ | $6.89 \pm 0.04$ | $1266.2 \pm 0.0$ | $24.1 \pm 0.1$ |
| Our DNGO | $0.432 \pm 0.18$ | $-3.229 \pm 0.18$ | $7.84 \pm 0.47$ | $1277.4 \pm 5.84$ | $24.1 \pm 0.0$ |
| BOHAMIANN | $0.399 \pm 0.001$ | $-3.223 \pm 0.18$ | $8.48 \pm 1.6$ | $1267.6 \pm 2.14$ | $24.1 \pm 0.0$ |

Figure 3: Comparison between GP-based optimizers and BOHAMIANN for multi-task Bayesian optimization for the WINE (for a random forest) and HEPATITIS dataset (using an svm). We plot mean $\pm$ 1 stddev of the distance to the optimum, based on 10 independent runs. Both multi-task capable approaches find good hyperparameters quickly.

### D.3 Details for the multi-task experiments

The 21 datasets from the OpenML project website [2] that we used are listed in Figure 4.

The hyperparameters of the SVM and random forests for this task are listed in Table 4.

Exemplary plots for multi-task Bayesian optimization on two data-sets are depicted in Figure 3

### D.4 Details for the deep reinforcement learning experiment

#### D.4.1 Experimental setup

In the RL experiments we trained DDPG using the continuous state information in the classical Cartpole task and a two link robot arm reaching task. For the Cartpole the dimensionality of the state was 4 (cart position, cart velocity, pole angle, pole velocity) and the one-dimensional action corresponded to motor torques (-10 to 10 Nm) applied to the cart. The goal in this task is to swing-up the pole from a resting position and balance it in the middle of the cart-track. For the reward function we penalized the agent with a large negative reward of $-10$ when it crashed into the walls at the end of the track and otherwise gave a reward proportional to the distance between the pole angle and cart position to their desired position (cart in the middle of the track and pole upright). Each episode lasts for 150 steps and we performed 50 test episodes in between every 5 training episodes.

For the two-link arm reaching task the state consists of the angles of the two joints and their veolicities, yielding a total of 6 state dimensions. The two dimensional action vector corresponds to applying forces to the motors of the two joints. The reward in this task is given as the euclidean distance between the joint positions and a target joint configuration (multiplied by 0.1) and a small penalty discouraging large actions.

| Dataset name | # Features | # Patterns | # Classes |
|---|---|---|---|
| **wine** | 14 | 178 | 3 |
| breast-w | 9 | 699 | 2 |
| vote | 16 | 435 | 2 |
| iris | 4 | 150 | 3 |
| monks-problems-1 | 7 | 556 | 2 |
| monks-problems-2 | 7 | 601 | 2 |
| heart-c | 13 | 303 | 2 |
| **heart-h** | 13 | 294 | 2 |
| heart-statlog | 13 | 270 | 2 |
| hepatitis | 19 | 155 | 2 |
| ionosphere | 34 | 351 | 2 |
| **hepatitis** | 19 | 155 | 2 |
| labor | 16 | 57 | 2 |
| lymph | 18 | 148 | 4 |
| sonar | 60 | 208 | 2 |
| **pendigits** | 16 | 10992 | 10 |
| mfeat-karhunen | 64 | 2000 | 10 |
| mfeat-morphological | 6 | 2000 | 10 |
| mfeat-zernike | 47 | 2000 | 10 |
| yeast | 8 | 1484 | 10 |
| vowel | 13 | 990 | 11 |

Figure 4: List of the datasets used for the multi-task experiments the target tasks of the four groups are marked in bold.

Table 4: Hyperparameters for the SVM and Random Forest in the multi-task experiment.

| Method | Hyperparameter | Values | # Values |
|---|---|---|---|
| SVM | $\log_2(C)$ | $\{-5, -4, \ldots, 15\}$ | 21 |
| SVM | $\log_2(\gamma)$ | $\{-15, -14, \ldots, 3\}$ | 19 |
| SVM | variance to keep | $\{80\%, 90\% \ 100\}$ | 3 |
| RF | min splits | $\{1, 2, 4, 7, 10\}$ | 5 |
| RF | max features | $\{1\%, 4\%, \ldots, 100\%\}$ | 10 |
| RF | criterion | $\{$Gini, Entropy$\}$ | 2 |
| RF | variance to keep | $\{80\%, 90\% \ 100\}$ | 3 |

### D.4.2 Evaluation metric

We posed the Bayesian optimization problem as minimizing the number of episodes (executed trajectories in the RL environment) required until the task is sucessfully solved in the 10 following test episodes (which are interleaved with the collection of new training episodes).

To solve both tasks we parameterized a fully-connected neural network for representing the Q function and a network for representing the policy $\pi$, yielding a total of 13 hyperparameters. All optimized hyperparameters are listed in Table 5.

The most important parameters were the learning rate and the ratio of the number of updates to the number of collected data-points (executed between training episodes). Additionally, BOHAMIANN consistently chose not to use batch normalization (as in the original paper) but applied weight normalization [8] to all layers of both networks – and did not employ any weight decay.

### D.5 Residual networks on CIFAR-10

As described in the main paper, we optimized the hyperparameters of a convolutional residual neural network for object recognition from visual images. This model obtains state-of-the-art performance on several image classification datasets (when the hyperparameters are set "correctly") and was used to win the ImageNet challenge in 2015. We tuned the parameters of the stochastic gradient descent procedure used to train these networks (including learning rate, the learning rate decay a momentum term and the number of epochs for training and a global weight decay). Additionally we parameterized several key architectural choices in the construction of the network. These include: the

Table 5: Hyperparameters for the deep RL experiments and their optimized settings.

| Hyperparameter | Range | Cartpole Optimum | Reaching-task Optimum |
|---|---|---|---|
| Learning Rate (LR) Q | $[10^{-8}, 0.1]$ | $10^{-3}$ | $10^{-3}$ |
| LR $\pi$ | $[10^{-8}, 0.1]$ | $10^{-4}$ | $10^{-3}$ |
| LR target networks | $[10^{-8}, 0.1]$ | $10^{-4}$ | $10^{-4}$ |
| Units Layer 1 Q | $[20, 500]$ | 120 | 150 |
| Units Layer 2 Q | $[20, 500]$ | 108 | 109 |
| Units Layer 1 $\pi$ | $[20, 500]$ | 80 | 94 |
| Units Layer 2 $\pi$ | $[20, 500]$ | 82 | 101 |
| Weight decay Q | $[0, 0.1]$ | $10^{-10}$ | $10^{-11}$ |
| Weight decay $\pi$ | $[0, 0.1]$ | $10^{-11}$ | $10^{-9}$ |
| Batch normalization | 0,1 | 0 | 0 |
| Weight normalization | 0,1 | 1 | 1 |
| Updates per step | $[1, 30]$ | 11 | 9 |

Figure 5: Validation Error as a function of total run-time for parallel Bayesian optimization (with 8 parallel function evaluations) for BOHAMIANN and DNGO. We show the validation error of the best currently found configuration (solid lines) and a scatter plot of all incoming results over time.

Table 6: Comparison of the 32 layer ResNet with hyperparameters optimized via BOHAMIANN and state-of-the-art results from the literature. Results marked with ‡ were taken from He et al. [9]. * denotes that this this result was obtained using a re-implementation of ResNet in theano [10].

| CIFAR-10 | Error (%) |
|---|---|
| DSN | 8.22 |
| ResNet‡ (32 layers) | 7.51 |
| ResNet‡ (1k layers) | **4.62** |
| ResNet* (32 layers) | $7.40 \pm 0.3$ |
| BOHAMIANN + ResNet-32 | $7.12 \pm 0.27$ |

strategy used for spatial dimensionality reduction, the form of the identity mapping used in residual networks and, additionally, we allowed for shared parameters between consecutive residual layers. A complete listing of the optimized hyperparemters, together with their ranges and optimized settings, is given in Table 7.

Table 7: Hyperparameters for the residual network experiment and their optimized settings for CIFAR-10. If percent shared $> 0$ it denotes the number of layers in each ResNet block that share their parameters (e.g. with percent shared $= 0.5$ the first 50% of the layers in each block share parameters).

| Hyperparameter | Range | Optimum |
|---|---|---|
| Learning Rate (LR) | $[10^{-5}, 0.9]$ | 0.197 |
| Learning rate decay | $[10^{-4}, 0.5]$ | 0.12 |
| Weight decay | $[10^{-8}, 0.1]$ | $3.16 \cdot 10^{-0.5}$ |
| 1 - Momentum | $[10^{-8}, 0.5]$ | 0.12 |
| percent shared | $[0, 1]$ | 0.5 |
| Epochs | $[60, 120]$ | 98 |
| Dim. reduction | ['strided', 'max-pool', 'average pool'] | max-pool |
| Number of blocks | $[0, 5]$ | 5 |

The results for optimizing this model on the CIFAR-10 [11] dataset using 8 parallel versions of both DNGO and BOHAMIANN are shown in Figure 5. To obtain these results we used the the parallel BO procedure outlined in Section A for this experiment, optimizing validation error. For validation, the last 10000 data-points from the training set were selected (and subsequently not used for training). All methods quickly find good configurations of the hyperparameters and BOHAMIANN reaches the validation performance of our manually-tuned baseline ResNet implementation after 104 function evaluations (or approximately 27 hours of total training time). A comparison between the resulting model (when re-trained on the full training data-set) and the vanilla ResNet is given in Table 6.

## Footnotes

[1]As is often the case with hyperpriors, we found our method to be insensitive to the exact choice of $\alpha$ and $\beta$.

[2]http://www.openml.org/