[Reviews · NeurIPS 2016]

Reviewer 1

Summary

This paper proposes to use neural networks as a powerful and scalable parametric model, while staying as close to a truly Bayesian treatment as possible. It derives a more robust variant of the recent stochastic gradient Hamiltonian Monte Carlo (SGHMC) method. Experimentations are showing the power of the method

Qualitative Assessment

The derivations are the main content of the paper, but we need more details on the Time complexity of the model. Please complete the results with CPU time and comparisons...

Confidence in this Review

2-Confident (read it all; understood it all reasonably well)


Reviewer 2

Summary

This paper develops a novel approach to Bayesian optimization using Bayesian deep neural networks. In particular, the authors combine recent advances in approximating Bayesian neural networks (stochastic gradient MCMC and Hamiltonian Monte Carlo) to be effective at Bayesian optimization. Bayesian optimization is traditionally performed using Gaussian processes, but this does not scale well and thus becomes computationally intractable for many observations (or few observations across many different tasks). The authors show that their approach usually outperforms the one existing formulation for Bayesian optimization with neural networks on a variety of optimization tasks.

Qualitative Assessment

It is very difficult to get most of the existing approaches to approximate Bayesian neural networks to work robustly without a significant amount of problem specific tuning (which makes them a poor choice for a task such as Bayesian optimization). I really appreciate the authors' commitment to making the methodology generalize across different problems. In general, I think this is an excellent paper. One glaring omission in this paper is any evaluation of the wall-clock runtime of the approach. How long does it really take in practice to perform 2000 burn-in steps and then 50*50 = 2500 steps for thinning and sampling? I'm curious how many data points one could reasonably accommodate while performing a parallel Bayesian optimization before the optimization procedure becomes the bottleneck. Having a latent vector representation for each experiment in the multitask setting seems quite sensible. However, I don't see much detail about how these latent vectors are learned. Are they learned in the same way as the model weights? What are the priors on these vectors? The choice of prior could certainly have a non-trivial impact on the performance in Bayesian optimization. e.g. setting the the prior to zero-mean Gaussian encodes a notion that all tasks are similar (i.e. all the zero-vector) in the absence of observations to the contrary. That seems like a sensible prior for this, but it would be nice to have a discussion about this. I disagree that a one-hot encoding would be less efficient; in fact I think it would be exactly the same. With a one hot coding you are essentially just treating the weights attached to each input as the latent vector which the one hot code just indexes into. This can be done as efficiently as this latent vector approach. The supplementaries contain some really interesting additional information. I appreciated the plots showing the sinc fit of various other approximations for Bayesian neural nets. I have done very similar experiments myself and definitely agree that the uncertainty estimates from these is not of high enough quality to use in Bayesian optimization. However, in the fit of the sinc function shown in Figure 1, you can see that the posterior mean of the function dips down as it gets farther from the data on the left side but up on the right side. This seems like a result of the highly non-stationary model and an undesirable quality for Bayesian optimization. For example, the optimization routine (assuming we are minimizing) would arbitrarily explore on the left side instead of the right side. This doesn't seem to be as much of an issue for the other approaches, as their means seem to asymptote towards the same value on either side. It would be nice to have an understanding of why this happens and why it might not be a problem. In the supplementaries it looks like DNGO outperforms SGHMCBO on at least one of the benchmarks (Hartmann 3). I'm curious why it seems to perform better on this problem. Does SGHMCBO perhaps do better as the dimensionality of the problem increases (and thus there are more model parameters)? Nit picky comments: Line 49: It seems weird to indicate the number of experiments using #i since i is a scalar. Why not follow convention and just use n, i.e. i=1...n

Confidence in this Review

3-Expert (read the paper in detail, know the area, quite certain of my opinion)


Reviewer 3

Summary

This paper presents a method for using Bayesian neural nets in the context of Bayesian optimization. It is similar in spirit to DNGO, but uses stochastic gradient HMC in place of a Gaussian approximation to the posterior. For efficiency, it uses a diagonal mass matrix which mimics algorithms like Adagrad or RMSProp. Experiments show significant improvements in predictive likelihood on benchmark datasets and modest improvements over DNGO in a Bayesian optimization setting.

Qualitative Assessment

Overall, I think this is a strong paper. It is quite well written, and clearly explains the relationships with existing methods and the novel contributions. The experiments are thorough, and improvements are demonstrated across a range of conditions. Even if the improvements over DNGO are modest, it's still a useful contribution to have an alternative approach which is practical in a Bayes opt setting. What are the computational requirements of the proposed approach? How does the per-evaluation time compare with DNGO? Is computation a limiting factor, i.e. would the BO results improve much if a lot more samples were generated? How well do other BNN approximations (e.g. Bayes by Backprop, dropout) work in the context of BO? Based on Figure 1 of the supplemental, it seems like they have some pathologies which would cause BO to fail; if so, maybe that's worth a few sentences. Minor comments/corrections: - the E[t] notation is confusing, since it looks like an expectation - in equation (7), I think the last term should be squared - in the notation mu(x | D), I think it should be mu(f(x) | D) - in "we then take a Gaussian approximation to this posterior", I think it should be "approximation to the predictive distribution" - M is not an "approximate preconditioner", it is a preconditioner which uses an approximation to the Fisher matrix - in Table 1, the baseline results aren't bolded, even when the baseline outperforms SGHMC - "in contrast to SGHMCBO, its performance is dependent on correctly setting all hyperparameters": I don't draw this conclusion from the figure - in Figure 1 of the supplemental material, include a plot for the DNGO approximation

Confidence in this Review

2-Confident (read it all; understood it all reasonably well)


Reviewer 4

Summary

This paper suggest a method to adaptively tune the parameters of Stochastic Gradient Hamiltonian Monte Carlo (SGHMC), and then use SGHMC to do to do Bayesian Optimization (BO). This method is then tested on several benchmarks and compared with previous approaches.

Qualitative Assessment

Adapting SGHMC (or any SG-based sampling based method) for BO seems like a very natural idea, and I'm happy this approach is being seriously explored in this paper. The authors performed quite a few numerical tests for this method, and got nice results, except the following issues which needs to be addressed: 1) The authors only compare with random search in the low dimensional cases. They should also compare with random search in high dimensional cases (Cifar10 and cartpole). It would be even better be to compare with more sophisticated methods, such as Hyperband. 2) The authors do compare their SGHMC method with their implementation of the DNGO method on the Cifar10 and the cartpole Benchmarks. However, in the first task it would be more convincing if the authors used the same network as in the DNGO paper (Snoek et al.), which got better performance (6.37%) then the network reported in this paper (7.12%). This difference may make the reader suspicious, since Table 3 in the supp. suggests this DNHO implementation is not performing as well as original implementation (in Snoek et al.). 3) Regarding the comparison with other Bayesian methods (VI,PBP, and dropout). It is not clear to me why does the uncertainty increases when we do extrapolation (going to regions far from the dataset) only for SGHMC, but not for the other methods (Figure 1 in supp.). This seems strange since this type of uncertainty does seem to increase in PBP paper (Figure 1 there). This point seems somewhat important, since the authors claim (section 2.2) this problem in extrapolation is main advantage of SGHMC over the other Bayesian methods (VI,PBP, and dropout), which seem to get good (and sometimes better) predictive performance (table 1 and 2 in supp.). Any clarification would be helpful. Minor comments: 1) In several cases, when the experimental results are mentioned in section 6, the relevant figures/table are not mentioned (e.g. first paragraph of 6.1, and also 6.3). 2) The order of the results in the figures and text in section is different, which is confusing. 3) The fonts in some figures are really tiny... 4) I think there is some confusion in the references (e.g., ref. [29] seems to be a combination of a title and authors from two different papers). %% After Rebuttal %% I appreciate the authors efforts to address my concerns. I have raised my scores, Due to the new results and explanation.

Confidence in this Review

2-Confident (read it all; understood it all reasonably well)


Reviewer 5

Summary

This paper fills an important niche, following on recent work showing that it was possible to train and sample Bayesian neural networks using stochastic updates similar to SGD (with adaptive momentum, here). This could be useful in many settings where estimating the model uncertainty is important, such as the demonstrated multi-task Bayesian optimization.

Qualitative Assessment

The overall procedure is fairly complex. Even with the appendix, I am not sure that it would be reproducible without quite a bit of fiddling. I will keep my reasonably good scores conditional on the authors promising to post their code online. I am surprised that the authors used a fairly shallow network (3 layers) with tanh units. Is it because a deeper net based on ReLUs did not work or because they did not try the latter? Would it make sense to introduce dropout in the training procedure? The number of burn-in steps should be specified in the main paper. How sensitive is that choice? Is that 2000 epochs or 2000 mini-batches? There is no discussion about the choice of C in the main text. What is it supposed to represent, ideally? l. 189 what are "all parameters"? v, \hat{V}_theta, \tau? table 1: what is the first line, "SGHMC (best average)" standing for? Clarify if the "SGHMC (scale-adapted)" is the "proposed" method. Fig. 2 is unreadable once printed. l. 268, clarify what "up to 0.25 to the minimum" means. In what units? typos: l.271, l.274 (missing 'to')

Confidence in this Review

3-Expert (read the paper in detail, know the area, quite certain of my opinion)


Reviewer 6

Summary

The use of vanilla Gaussian Processes for optimization problems that do not admit a natural loss function (such as optimization of hyperparameters), does not scale well to large data-sets and high-dimensional domains. However, these are typically the regions of interest in many practical applications. In particular, one has to iteratively evaluate an acquisition function to determine where to acquire new samples of the target function, but this is untractable since marginalization over the model parameters is untractable when the probabilistic model is in a non-trivial (parametric) model class. The paper proposes 1) to use neural networks to model the mean of the GP 2) to use a scale adapted version of the Hamiltonian MC sampling method to obtain a sampling estimate of the posterior. This sampling method simulates a Hamiltonian dynamical system to obtain samples. The spirit of this adaptation is similar to AdaGrad: HMC includes a "mass" that acts as a scaling of the "learning-rate". The authors rescale this "mass" using a variance estimate of the gradient of the log likelihood. This is only done during the burn-in phase of the MC procedure. 3) to adaptively set the exponential discounting window for computing the gradient variance, using the gradient and gradient variance themselves. In the experimental results, the authors demonstrate in 1 synthetic and 3 common problem settings (ResNet and DeepRL optimization) that this rescaling scheme finds the optimal hyperparameters faster than some current GP approaches, although the solutions found are not "better".

Qualitative Assessment

The authors claim that their method is more robust and scales well to larger problems / models. The speed-up results are promising, as they show non-trivial speed-ups across different experiments. 1) How hard is this to make work? Robustness means that a reasonable range of hyperparameter settings (i.e. epsilon and C) give good performance of SGHMCBO for different domains and problems. For SGHMCBO, the existence of a good choice is suggested by the experiments. However, how sensitive is the robustness to changing epsilon, C? Does the performance of SGHMCBO change similarly across all experiments, or are the values found by the authors the ones where SGHMCBO happens to perform well for these experiments? 2) The scale of the datasets is relatively "small", with CIFAR being the largest (~50k datapoints) - all others seem much smaller. The paper does not discuss or evaluate the use of their method on ImageNet-size (or larger) datasets -- how can we expect this method to perform in that regime? It is unclear to me from the results or the conceptual development in this paper how this method extrapolates to larger datasets. 3) What is the impact of the use of neural networks when using SGHMCBO? What happens when one uses a different model class? It would be desirable to see some experiments with a different model class used.

Confidence in this Review

2-Confident (read it all; understood it all reasonably well)